# Retrograde Intrarenal Surgery for Lithiasis Using Suctioning Devices: A Shift in Paradigm?

**DOI:** 10.3390/jcm13092493

**Published:** 2024-04-24

**Authors:** Petrisor Geavlete, Razvan Multescu, Cristian Mares, Bogdan Buzescu, Valentin Iordache, Bogdan Geavlete

**Affiliations:** 1Department of Urology, “Saint John” Emergency Clinical Hospital, 042122 Bucharest, Romania; geavlete@gmail.com (P.G.); pc_valii@yahoo.com (V.I.); bogdan_geavlete@yahoo.com (B.G.); 2Faculty of General Medicine, University of Medicine and Pharmacy, ‘Carol Davila’ Bucharest, 050474 Bucharest, Romania

**Keywords:** aspiration, direct in-scope suction, flexible ureteroscopy, negative pressure, steerable catheters, vacuum-assisted ureteral access sheath

## Abstract

New suction endoscopes, ureteral access sheaths (UAS) and catheters aim to improve the efficacy of flexible ureteroscopy and optimize its safety. Suction UAS with non-flexible tips have shown promising results, especially in maintaining low intrarenal pressure, but also in removing small debris and reducing the “snow globe” effect. In addition, suctioning UAS with a flexible tip offers the advantage of being able to be navigated through the pyelocaliceal system to where the laser lithotripsy is performed. It can also remove small stone fragments when the flexible ureteroscope is retracted, using the Venturi effect. Direct in-scope suction (DISS) involves aspirating dust and small stone debris through the working channel of a flexible ureteroscope, thus regulating intrarenal pressure and improving visibility. Steerable aspiration catheters are other devices designed to increase stone clearance of the pyelocaliceal system. They are inserted under fluoroscopic guidance into every calyx after retraction of the flexible ureteroscope, alternating irrigation and aspiration to remove dust and small gravels. Combining flexible-tip suction UAS and the DISS technique may offer some advantages worth evaluating. The advantage of using these instruments to achieve a low intrarenal pressure was demonstrated. The true practical impact on the long-term stone-free status is a matter requiring further studies.

## 1. Introduction

UAS is nowadays a frequently employed accessory during flexible ureteroscopy, but it is also a subject of controversy. The routine use of such an instrument as well as stenting habits are probably the two most debated issues regarding this type of procedure.

Easy repetitive access to the upper urinary tract but especially maintaining a low intrarenal pressure are probably the most important arguments of urologists advocating for the routine use of UAS. Preventing irrigation-induced increased pressure in the pyelocaliceal system reduces the risk for pyelovenous and pyelolymphatic reflux, ultimately translating into a lower rate of septic complications.

The management of small stone fragments and debris resulting from laser lithotripsy is another issue clearly requiring improvements ever since the early period of the flexible ureteroscopic approach. With the development and improvement of lasers and flexible endoscopes, the stones treated by such methods become larger, generating more stone material during lithotripsy and consequently making this matter even more significant.

## 2. Literature Search

Literature was searched for significant articles on suctioning devices (Figure 1).

Pubmed was searched using the query “((suctioning ureteral access sheaths) OR (vacuum assisted ureteral access sheaths)) AND ((flexible ureteroscopy) OR (flexible ureterorenoscopy) OR (retrograde intrarenal surgery))”. After eliminating from the search results letters to the editor, systematic reviews, and articles without relevance to the searched topic, 21 articles remained for analysis: 9 for the suction UAS with standard (non-flexible) tip section and 12 for the suction UAS with flexible-tip section. Because one article described only the use of suction through a flexible-tip UAS and not the insertion in the calices (steering capabilities), the article was discussed in the section dedicated to UAS with a non-flexible tip. After content evaluation, one article analysis was included in the section dedicated to direct in-scope suction.

Another Pubmed search was performed using the query “(flexible ureteroscopy) AND (DISS OR direct in-scope suction)”, after which 3 studies were included in the evaluation.

Regarding the steerable catheter, Pubmed was searched using the query “(steerable catheter OR sure) AND (ureteroscopy)”. After relevance evaluation, 2 articles were included in the review.

Finally, we performed an internet search on producer sites for devices matching the definition in each section.

## 3. New Devices with Suction Capabilities

The expectations for every new accessory device are to improve the efficacy of the method (probably one of the most revealing parameters is the largest stone volume that is reasonable to be treated by it) and to optimize its safety. However, each of the suctioning devices came with certain aims and goals and has its own strong points that we will review in the following sections.

### 3.1. Suctioning UAS with Standard (Non-Flexible) Tip

This type of access sheath should be placed with the distal tip at the level of the uretero-pelvic junction or in the renal pelvis. Its main two advantages seem to be keeping a low intrapelvic pressure during lithotripsy and achieving effective removal of small-size stone particles and debris, thus improving visibility. Synonyms found in articles for this device are negative pressure UAS or vacuum-assisted UAS.

Three early articles reported promising initial results with suction UAS: one on a porcine model, evaluating changes in intrarenal pressure when using such a device [1], one using a standard UAS, which was modified to allow suction [2], and a third one employing a patented model of suction UAS [3].

Another three studies evaluated the correlation between irrigation flow and intrarenal pressure while using different types of flexible ureteroscopes and suction UAS. One of these studies, by Han Z et al., using an ex-vivo porcine kidney model, found that when standard UAS was used, intrarenal pressure of 30 mmHg was reached at an irrigation fluid flow of 60–70 cc/min. This is close to the cut-off value of 35–40 mmHg when the pyelotubular back-flow usually occurs [4,5]. By comparison, when a suctioning UAS was used with the lateral vent closed, the intrarenal pressure stayed around 10 mmHg, even at very high irrigation fluid flow rates of 120 cc/min. The vent is a small opening on the aspiration port of the UAS, which can be opened or closed; the latter instance allows UAS to work as a closed vacuum system [5]. In a similar fashion, Guan et al. found that when using 12/14F and 11/13F vacuum-assisted UAS with the vents closed, the intrarenal pressure remained at an encouraging level of less than 5 mmHg at high levels, with irrigation fluid flow rates of 200 cc/min [6].

The size of the stone fragments and debris which are adequate for expulsion (small enough to be aspirated) was the subject of one study. This dimension seems to double when suction (negative pressure) UAS are used, in comparison to conventional ones [7].

Zhu et al., using an early model of suction access sheath, completed the flexible ureteroscopic approach by adding a supplementary operation. At the end of the procedure, after the retraction of the flexible ureteroscope, they inserted a 5F catheter through the access sheaths and injected saline for 20–40 s, thus generating a circuit of irrigation liquid to further wash the pyelocaliceal system. The authors described shorter operative times, improved immediate postoperative stone-free rates, and lower complication rates in patients treated using the suction UAS. This group presented significantly lower incidences of fever and urosepsis in comparison to the standard UAS group. However, the one-month stone-free rate was similar between the two groups. The authors hypothesized that the device has the potential to at least reduce the amount of time the double-J stent is indwelled [8].

One study, despite employing a flexible-tip UAS, appears to use only the suctioning capabilities of the sheath but not of the intrarenal steering, so, as stated in the Literature Search section, we decided to index it in this chapter. The authors compare a flexible ureteroscopic approach using this accessory with minimally-invasive percutaneous nephrolithotomy for 1–2 cm upper-ureteral infectious stones, relocated in the renal pelvis or the superior or middle calices. By comparison to the percutaneous access, the retrograde approach presented a statistically significant better immediate stone-free rate, and similar stone-free rates at two weeks and one month. Predictably, retrograde ureteroscopy was associated with a shorter hospital stay and a lower incidence of fever, pain and urosepsis [4].

Huang et al. evaluated suction UAS in patients with pyelocaliceal stones on solitary kidneys, reporting good efficacy and safety [9].

One case report described the employment of vacuum-assisted UAS using an improvised system: at the end of the laser lithotripsy, a smaller UAS was inserted through the preexisting larger one and used for stone fragment extraction by suction [10].

Most of the data evaluated in this section were reported by groups from China. One case was reported by authors from the USA and one study was reported by authors from both China and the USA.

### 3.2. Flexible-Tip Suctioning UAS

While the models of suctioning UAS described in the previous section must be placed at the level of the uretero-pelvic junction, the flexible-tip models offer the advantage that they can be navigated through the pyelocaliceal system, including into the inferior calyx. Other terms used to indicate this type of UAS that can be found in different articles are distal active flexible UAS, flexible and navigable UAS, bendable-tip UAS, or omnidirectional UAS.

Examples of commercially available suction bendable UAS described in the following articles are ClearPetra (Well Lead Medical Co, Guangzhou, China) or Elephant II (Zhejiang YiGao Medical Technology Co., Ltd., Hangzhou, China).

The flexible tip of such UAS may offer significant maneuverability. In a study by Ding et al., the deflection angle of the flexible UAS varied between 110–130 degrees with an empty working channel and 90–130 degrees with various accessory instruments inserted [11].

There are different techniques described for small stone fragments and debris removal using this device. Very small such fragments positioned in a space within 10 mm from the tip of the sheath can be suctioned through the space between the ureteroscope and the sheath. Larger fragments (though obviously not larger than the inner diameter of the UAS) can be extracted using the suction while the flexible ureteroscope is slowly withdrawn [12]. This is based on the Venturi effect. The effect was described by the 18th-century Italian physicist Giovanni Battista Venturi and consists of the reduction of fluid pressure when it speeds up as it flows through a constricted section of a conduct.

Gauhar et al. evaluated the efficacy of a flexible ureteroscopic approach to renal stones using two combinations of flexible ureteroscopes and flexible UAS. The 10F flexible UAS combined with 7.5F ureteroscopes achieved better stone-free rates than the 12F flexible UAS with 8F scopes, with a probable explanation that the thinner UAS proved better flexibility. The study reported an excellent stone-free rate by comparison to studies employing standard UAS [13].

One issue is how intensely the vacuum should be applied through the suctioning UAS. A study by Ostergar et al. evaluated the influence of ClearPetra flexibile-tip suctioning UAS over intrarenal pressure in an ex vivo porcine model. Although the authors state that the distance between the sheath tip and the uretero-pelvic junction was one of the parameters evaluated, no information regarding its influence over the pressure was reported. The interesting finding is that the vacuum-assisted UAS may prevent the increase in intrarenal pressure, but this effect is reversed when a high vacuum rate (more than 200 mmHg) is applied, especially after 5 s, probably due to the collapse of the outflow tract. The study suggests using lower vacuum settings at short bursts of about 5 s [14].

Two studies compared flexible ureteroscopy using a flexible-tip UAS with the same procedure using the standard one, and in all of them, both the immediate and the one-month stone-free rates were better for the flexible-tip UAS group [12,15]. In one of the studies, significant differences were recorded, even for larger stones of between 2 and 3 cm [15].

Regarding safety, Zhang et al., comparing the results of the flexible ureteroscopic approach using flexible-tip suctioning UAS in 102 patients and the standard model in 112 cases, found no differences regarding intraoperative complication rates or hospitalization time. However, in the flexible-tip UAS, the operative time and stone-free rates, as well as the overall complication rates, were significantly lower. Flexible-tip suctioning UAS proved superior to the standard one in preventing septic complications and hemoglobin drop [12].

One of the most important effects of using even the conventional UAS is protection against septic complications by preventing intrarenal pressure increase. Qian et al. demonstrated in a study that vacuum-assisted UAS may reduce the incidence of postoperative SIRS (Systemic Inflammatory Response Syndrome) [16].

Two groups compared the flexible ureteroscopic approach using a flexible-tip UAS with minimally invasive percutaneous nephrolithotomy for treating larger stones between 2 and 3 cm [17,18]. As expected, the retrograde technique was associated with less hemoglobin drop, shorter postoperative stay, and lower overall complications, bleeding, and pain rates. The ureteroscopic method paired with the novel UAS offered statistically similar stone-free rates, both immediately and at one month postoperatively, thus offering a solid alternative even for larger stones [17].

Also, two case report articles described the complete removal of renal stones, one of them including multiple large ones [19,20].

Eight of the studies described in this section originated from groups from China, one from a group from the USA, and one from a group including authors from the USA and China. One study was performed in two centers and reported by a multinational group comprising authors from Singapore, France, Italy, India, Canada, the UK, and Saudi Arabia.

### 3.3. Direct In-Scope Suction

DISS refers to the aspiration of dust and small stone debris through the working channel of the flexible ureteroscope.

Deng et al. reported in 2016 the first results using a patented system comprised of a flexible ureteroscope with both irrigation and suctioning capabilities and a ureteral access sheath with a pressure sensor in the tip. The aim of the system was to regulate the intrarenal pressure by balancing the irrigation flow and the vacuum suctioning. The results were promising, with stone-free rates of 90% and 95.6% at one and 30 days, respectively. During the procedure, the pressure in the renal pelvis was kept under 20 mmHg, while the authors reported clear vision during the interventions [3].

Gauhar et al. described the initial experience with a cost-effective modification that may add direct in-scope suction to any semirigid or flexible ureteroscope. The accessory device was manufactured by connecting two 3-way stoppers, the two inlets being used to connect suction and irrigation, respectively. It was used together with a Flex-X2 Storz reusable flexible ureteroscope (Karl Storz, Tuttlingen, Germany) and a single-use Uscope flexible ureteroscope (Zhuhai Pusen Medical Technology Co., Zhuhai, China). The authors evaluated 30 patients who underwent this technique and, interestingly, compared them to 28 patients in which RIRS was performed using a suction flexible-tip UAS and the single-use Uscope flexible ureteroscope. Both methods proved safe and effective; however, the DISS group was associated with longer operative times and included significantly more cases (almost ten-fold) which, during follow-up, required additional procedures. The hospital stay was shorter in in-scope suction patients [21].

Kritsing et al. reported a clinical case in which a large ureteral calculus in a transplanted kidney was treated using a regular 8.4F Scivita flexible ureteroscope (Scivita Medical, Suzhou, China) with a similar improvised add-on device allowing alternative irrigation and in-scope suction. Applying the dusting technique, the authors were able to remove the entire stone and ensure a smooth postoperative recovery [22].

The papers evaluated in this section were authored by three groups: one from Thailand, one from China, and a multinational one from Singapore, the UK, Canada, Turkey, China, Italy, India, and France.

### 3.4. Steerable Aspiration Catheters

Another type of accessory instrument designed to improve intraoperative debris and stone fragment evacuation is the steerable catheter, such as the CVAC Aspiration System (Calyxo, Inc., Pleasanton, CA, USA), which is used to perform a procedure that was coined as SURE—Steerable Ureteroscopic Renal Evacuation. After flexible ureteroscopy is finished, the endoscope is retracted and the steerable catheter is inserted and navigated under fluoroscopic guidance into every calyx where irrigation and aspiration are alternatively applied. An initial report by Sur et al. suggests that SURE is a safe and feasible method, with better one-month stone-free rates than basket removal of stone fragments, although the difference was not clinically significant [23]. A multi-institutional center study demonstrated also the safety and efficacy of the method. A promising conclusion of the study was that among the patients anticipated to require multiple-staged procedures, only a small proportion (under 10%) actually required a second one. Among high-risk patients, the stone clearance reached very good rates, with no method-related complications [24].

One study described in this section was reported by a group from the USA and the other by a group of authors from the USA and India.

## 4. Discussion

All the types of endoscopes, UAS and steerable catheters with suction capabilities described above seem promising in regard to the improvement of efficacy and safety (Table 1).

In the evolution of these new types of UAS, the non-flexible tip models seem rather characteristic for the early stages, offering mainly the advantage of a procedure performed in conditions of low intrarenal pressure. Another secondary benefit is the at least partial removal of dust and debris from the renal pelvis, thus diminishing the “snowstorm” (or “stone globe”) effect.

Recent articles describe more frequently the experience with the bendable, flexible-tip UAS that can be steered and inserted into the calices, where the actual lithotripsy is performed. This type of device appears to be the next step in the evolution of UAS. With them, the suction of larger stone gravels becomes possible, with the potential to change the paradigm in the management of dust and debris resulting from laser fragmentation. This offers the possibility of a new “real stone-free” status over the previous “successful procedure” status. In the latter situation, the presence at the end of the procedure of stone fragments under 2–4 mm (with great potential of elimination in the next weeks, probably with no clinical impact) was considered absolutely acceptable. However, if this new status has a real significant practical impact in the long-term is a matter that probably requires further studies.

Regarding in-scope suction, all the flexible ureteroscope models used in the evaluated articles in their respective sections have a standard 3.6F working channel. Of course, only smaller-size debris can be suctioned through this channel, in comparison to the method employing suction UAS with larger lumen. Nowadays, new models of flexible ureteroscopes with built-in in-scope suction are commercially available, which are more ergonomic to use than the in-house modified ones. While the advantage of maintaining low intrarenal pressure is palpable, the real impact on the efficacy of the procedure also requires further studies.

Routine use of these devices may increase the instrumental costs of a flexible ureteroscopic approach but also has the potential to decrease the overall costs by reducing the need for supplementary procedures, decreasing stenting time, or preventing morbidity. The evaluated studies do not report in a systematic manner data regarding the correlation between the use of different suction devices and parameters such as vulnerable groups, economic background, etc. Such information was rarely reported, so no significant conclusions can be drawn in this regard.

## 5. Conclusions

Suction-capable accessories and endoscopes for flexible ureteroscopic approaches have been used on an increasing scale in the last years. They can lower the pyelocaliceal pressure during intrarenal procedures, increasing safety, especially by preventing septic complications. These instruments have the potential to improve visibility and small stone fragments clearance during RIRS, thus improving its efficacy. However, the true practical impact over the medium- and long-term success rates of the procedure is still a matter to be further studied.

## 6. Future Directions

Using combined flexible-tip suction UAS and DISS techniques may offer some advantages and is a matter worth being evaluated. In such procedures, DISS has the potential to improve stone fragment extraction through the UAS lumen by augmenting the Venturi effect. The study by Gauhar et al. used a Uscope single-use ureteroscope paired with a flexible-tip suction UAS as a comparator to the DISS group. However, the authors do not mention if the suction capability of the flexible ureteroscope was also used in this group [21].

Taking into consideration that more such suction devices are now commercially available from different manufacturers and there is the possibility to combine them to obtain maximum effects, there is room for significant evolution, with the potential to reshape the retrograde flexible ureteroscopic approach of the upper urinary tract.

## Figures and Tables

**Figure 1 jcm-13-02493-f001:**
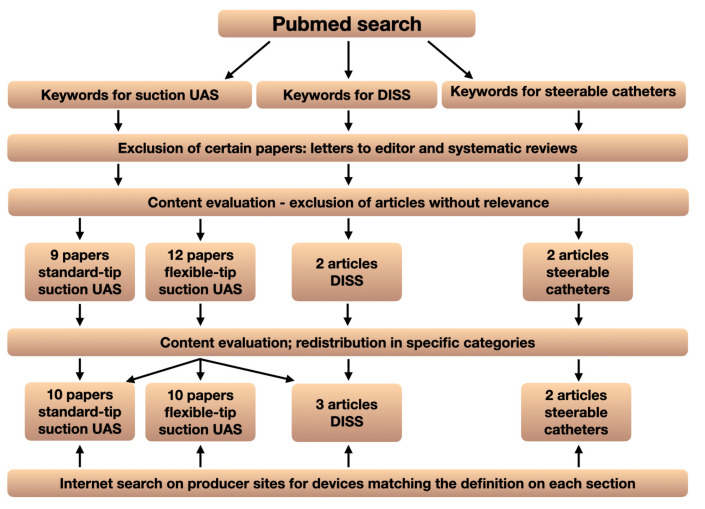
Literature search and the selection process of the papers to be evaluated in the review.

**Table 1 jcm-13-02493-t001:** Comparative view of the main characteristics of some of the evaluated clinical studies. Due to the heterogeneous data reported, some of the clinical studies were not included in this table. Stone-free rates were evaluated on the first postoperative day (SFR1), after 30 days (SFR30), and after 90 days (SFR90).

Author	Number of Renal Units	Stone Size	Operative Time (min)	SFR1	SFR30	SFR90	Complications Rate
**Standard tip suction UAS**							
Deng X, 2016 [3]	90	15.9 ± 5.2 mm (8–35 mm)	24.8 ± 15.9 (13–49)	90.0%	95.6%	N/A	16.7% (14.5% Clavien I and 2.2% Clavien II)
Tang QL, 2023 [4]	86	16 ± 4 mm	61.4 ± 5.2	73.2%	94.2%	N/A	6%
Zhu Z, 2019 [8]	165	18.2 ± 5.2 (8–35) mm	49.7 ± 16.3	82.4%	88.8%	N/A	11.5% (8.5% Clavien I, 1.8% Clavien II, 0.6%, Clavien III, 0.6% Clavien IV)
Huang J, 2018 [9]	40 *	9–30 mm	25.2 ± 14.5	N/A	87.5%	92.5%	5% (all Clavien I)
Tapiero S, 2019 [10]	1	12 × 8 × 13 mm	N/A	100%	N/A	N/A	0%
**Flexible tip suction UAS**							
Ding J, 2023 [11]	153	13.0 ± 6.9 mm	42.0 ± 13.2	N/A	94.2%	N/A	0% major or septic complications
Zhang Z, 2023 [12]	102	18.5 ± 4.7 mm	55.3 ± 11.4	86.3%	91.2%	N/A	11.8% (5.9% Clavien I, 3.9% Clavien II, 1% Clavien IV, 8.8% septic complications)
Gauhar V, 2023 [13]	16 ^#^	21 (17–24.3) mm	63 (52–74.5)	N/A	N/A	68.8% ^&^	6.3% (Clavien I)
	19 ^$^	19 (12–22) mm	76 (63–82.3)	N/A	N/A	94.7% ^&^	10.5% (5.3% Clavien I, 5.3% Clavien II)
Huang J, 2023 [15]	103	<20 mm	37.7 ± 20.1	86.7%	96%	N/A	
		20–30 mm		57.1%	89.3%	N/A	
Qian X, 2022 [16]	81	19 (16–22) mm	72.9 ± 28.1	86.4%	88.9%	N/A	3.7% fever, 1.23% SIRS
Chen H, 2019 [18]	46	20 to 30 mm	65.6 ± 22.5	N/A	93.1%	N/A	11.3% (all Clavien I)
Yue G, 2023 [19]	1	10.5 × 10.4 mm	N/A	100%	100%	100%	0%
Xiao J, 2024 [20]	2	32 × 21 mm and 18 × 12 mm	N/A	100%	100%	100%	0%
**DISS**							
Gauhar V, 2022 [21]	30	22 (18–28.8) mm	80 (60–100)	N/A	66.7% ^@^	N/A	36.7% (Clavien I)
Kritsing S, 2024 [22]	1 ˜	20 mm	90	100%	100%	100%	0%
**Steerable suction catheter**							
Sur RL, 2022 [23]	11	267 mm^3^	54 ± 17 (6–35.9)	N/A	100%	N/A	11.1% (all Clavien I)

* Solitary kidneys; ^#^ First generation 10F Elephant II UAS; ^$^ Second generation 12F Elephant II UAS; ^&^ In all cases that were not stone-free, the residual fragments were smaller than 4 mm; ^@^ Evaluation at 21 days instead of 30 days; ˜ Large ureteral calculus in patient with transplanted kidney.

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
