# Peer review of "Retrograde Intrarenal Surgery for Lithiasis Using Suctioning Devices: A Shift in Paradigm?"

_jcm, 2024, doi:10.3390/jcm13092493_

Round 1
Reviewer 1 Report
Comments and Suggestions for Authors
The title of this review is: "Retrograde intrarenal surgery for lithiasis using suctioning devices: a shift in paradigm?"
Authors analized 21 articles regarding suction ureteral access sheaths with or without flexible-tip, 3 studies treating the direct in-scope suction technology and 2 articles regarding steerable aspiration catheters.
Authors concluded that suction accessories and endoscopes for flexible ureteroscopic approach are used on an increasing scale in the last years. They can lower the pyelocaliceal pressure during intrarenal procedures, increasing their safety, especially by preventing septic complications improving visibility. However, the true practical impact over the medium and long-time success rates of the procedure is still a matter to be further studied.
Par 3.1
Line 14.. coorect.. This is close to the cut off value of 35-40 mmHg when the pyelo-tubulor back-flow usually occurs [4, 5]
Author Response
We thank the reviewer for the input and suggestion. The text was corrected: ”the pyelo-tubulor“ was replaced with “the pyelo-tubular“. Changes were highlighted in text.
Reviewer 2 Report
Comments and Suggestions for Authors
The authors have performed a comprehensive study of all available suction devices to perform ureteroscopy and have highlighted the pros/cons of each technique. As the use of different types of Ureteral access sheaths(UAS) continues to grow, this study is significant as it will help clinicians make future decisions for the best treatment options for their patients. I have the following suggestions for the authors to strengthen the quality of their manuscript:
1. I suggest making a TABLE listing all the the devices in section 3 and listing the pros/cons, number of patients studied, size of stone, operative time required, complications and final conclusion. This table will be a nice over view and will add value to the paper.
2. For the literature search section a nice flowchart will be helpful that takes us through the pipeline employed to arrive at the current data.
3. Can the authors based on their literature search comment on how these techniques vary based on country/Continent?
4. Also it might be useful to list the vulnerable populations ( age/sex/economic background) for each surgery type if available. That is is one kind of suction device preferred method for a particular age group/gender or does that not matter /vary ?
Author Response
1. The authors have performed a comprehensive study of all available suction devices to perform ureteroscopy and have highlighted the pros/cons of each technique. As the use of different types of Ureteral access sheaths(UAS) continues to grow, this study is significant as it will help clinicians make future decisions for the best treatment options for their patients. I have the following suggestions for the authors to strengthen the quality of their manuscript:
Response: We thank the reviewer for the kind appreciation and especially for offering suggestions that will improve the quality and strength of our paper.
2. I suggest making a TABLE listing all the the devices in section 3 and listing the pros/cons, number of patients studied, size of stone, operative time required, complications and final conclusion. This table will be a nice over view and will add value to the paper.
Response: We want to thank the reviewer for the valuable suggestion. We agree that this table will be a nice overview improving the reader experience. We inserted a Table indexing most of the papers which evaluate the clinical use of suctioning devices.
For the literature search section a nice flowchart will be helpful that takes us through the pipeline employed to arrive at the current data.
Response: We inserted in the text a flowchart describing the selection process of papers. We totally agree that it will help the readers to asses the selection process that finally generated the data included in the paper.
3. Can the authors based on their literature search comment on how these techniques vary based on country/Continent?
Response: Thank you for your suggestion. The following phrases were added to the article:
Section 3.1 "Most of the data evaluated in this section were reported by groups from China. One case was reported by authors from USA and one study was reported by authors from both China and USA.”
Section 3.2 "Eight of the studies described in this section originated from groups from China, one from a group from USA and one from a group including authors from USA and China. One study was performed in two centres and reported by a multinational group comprising authors from Singapore, France, Italy, India, Canada, UK and Saudi Arabia.”
Section 3.3 "The papers evaluated in this section were authored by a group from Thailand, one from China and a multinational one from Singapore, UK, Canada, Turkey, China, Italy, India and France."
Section 3.4 "One study described in this section was reported by a group from USA and the other by a group of authors from USA and India.”
Changes were highlighted in text.
4. Also it might be useful to list the vulnerable populations ( age/sex/economic background) for each surgery type if available. That is is one kind of suction device preferred method for a particular age group/gender or does that not matter /vary ?
Response: This are indeed very interesting issues to discuss. However, no specific informations regarding them were included in most of the evaluated studies. The retrograde flexible ureteroscopic approach is a method that can be applied in any age or gender group and the suctioning devices aim to generally improve its results and safety. Because of this, probably there is no significant variation in results to be expected in this matter. The following comment was added at the end of Discussion section: "Routine use of this devices may increase the instrumental costs of flexible ureteroscopic approach but also have the potential to decrease the overall costs by reducing the need for supplementary procedure, decreasing stenting time or preventing morbidity. The evaluated studies don't report in a systematic manner data regarding correlation between the use of different suction devices and parameters such as vulnerable groups, economic background etc. Such informations were rarely reported so no significant conclusions can be drawn in this regard.” Changes were highlighted in text.